# Assessing the Ecotoxicity of Copper and Polycyclic Aromatic Hydrocarbons: Comparison of Effects on *Paracentrotus lividus* and *Botryllus schlosseri*, as Alternative Bioassay Methods

**Chiara Gregorin** [1,2]**, Luisa Albarano** [3,4]**, Emanuele Somma** [1,5]**, Maria Costantini** [3,]*** **and Valerio Zupo** [1,]***

1 Department of Marine Biotechnology, Stazione Zoologica Anton Dohrn, Villa Dohrn, Punta San Pietro, 80077 Naples, Italy; chiara.gregorin@szn.it (C.G.); emanuele.somma@szn.it (E.S.)
2 Department of Life and Environmental Sciences, Polytechnic University of Marche, Via Brecce Bianche, 60131 Ancona, Italy
3 Department of Marine Biotechnology, Stazione Zoologica Anton Dohrn, Villa Comunale, 80121 Naples, Italy; luisa.albarano@szn.it
4 Department of Biology, University of Naples Federico II, Complesso Universitario di Monte Sant'Angelo, Via Cinthia 21, 80126 Naples, Italy
5 Department of Life Science (DSV), University of Trieste, Via Giorgieri 10, 34127 Trieste, Italy
* Correspondence: maria.costantini@szn.it (M.C.); valerio.zupo@szn.it (V.Z.)

**Abstract:** Adult sea urchins and their embryos are ideal targets to investigate the medium- and long-term effects of various toxic agents, such as organic and inorganic pollutants, to forecast and mitigate their environmental effects. Similarly, small colonial tunicates such as Botryllid ascidians may reveal acute toxicity processes and permit quick responses for the management of contaminants impacting coastal waters, to preserve the functional integrity of marine ecosystems. This investigation compares the functional responses of two model invertebrates, the sea urchin *Paracentrotus lividus* and the sea squirt *Botryllus schlosseri*, to chronic and acute exposures to organic and inorganic toxic agents. Such heavy metals as copper produce both acute and chronic effects on marine biota, while polycyclic aromatic hydrocarbons (PAHs) mainly produce chronic effects at the concentrations ordinarily measured in marine coastal waters. Both models were tested over a range of concentrations of copper and PAHs. Copper triggered a clear effect in both species, producing a delay in the embryo development of *P. lividus* and a rapid death of sea squirts. *B. schlosseri* was less sensitive to PAHs than *P. lividus*. The results on both species may synergistically contribute to assess the toxicity of organic and inorganic compounds at various concentrations and different physiologic levels.

**Keywords:** model organism; rapid assessment; heavy metals; PAH; allochemicals; sea urchin; sea squirt; seawater

## 1. Introduction

Remediation techniques and strategies for the management of ecotoxicity events require a rapid and precise assessment of contaminants and a clear understanding of their interactions with the marine biota [1]. Adopting effective and rapid toxicity assessment methods is essential to facilitate timely answers and the correct management of exposure events of marine organisms to chemical contaminants, preserving the integrity of ecosystems [2]. Various animal and plant models contribute to the assessment of contaminants in aquatic ecosystems and permit the meeting of rigorous regulatory requirements [3]. Pollutants may even accumulate in the sediments [4] and may be re-suspended in the water column by stochastic perturbative events [5]. In addition, various contaminants alter the integrity of ecosystem assemblages, being biomagnified in marine organisms [6,7], and finally affecting human health [8–10]. To this end, the Water Framework Directive (WFD 2000/60/EU) and the Marine Strategy Framework Directive (MSFD 2008/56/CE) indicate coastal benthic communities as key elements to evaluate the impacts of pollutants on the

marine environment and suggest using benthic model organisms to perform toxicological tests of anthropogenically produced compounds. Echinoderms and tunicates are ecologically important organisms, widely distributed in aquatic environments, and they may be suitable models for aquatic toxicity tests. In particular, sea urchins (mainly, *Paracentrotus lividus* and *Arbacia lixula*) are often used for toxicity assessments. Their embryos permit the rapid assessment of acute toxicity events according to formal techniques [11], while adult individuals may be used for the evaluation of chronic effects of chemical pollutants [5]. Ascidians (Chordata, Tunicata, Ascidiacea), in their turn, are sessile filter-feeders extensively selected as models for various bioassays [12,13], including physiology and ecology tests [14], genetics [15], developmental biology [16], and toxicology [13,17].

In particular, the "golden-star" tunicate *Botryllus schlosseri* Pallas (1766) is a cosmopolitan colonial ascidian (Urochordata; [18]) recognized as an invasive species in the temperate seas of both hemispheres [19]. It is widespread [18], easy to collect [20], and the techniques for its laboratory rearing are known [14]. Its colonies exhibit both sexual and asexual reproduction [21]. Thus, clones produced by blastogenesis are available, as well as individuals expressing genetic variability, obtained by cross-fertilization [22]. Its larvae may be easily collected and colonies may be cultured for years. In addition, *B. schlosseri* exhibits the closest phylogenetic relation to vertebrates [23,24]. Its embryos and larvae share various homologies with vertebrates (e.g., the notochord and the dorsal neural tube) and express important developmental regulatory genes [20]. In this view, they are valid alternatives to vertebrates and could replace, reduce, or refine the use of fish in ecotoxicity tests. The ability of *B. schlosseri* to colonize new environments is one of the reasons pushing research to focus on this species, with the aim of rearing it in laboratory cultures and, thus, to have the possibility to investigate through a range of scientific fields [14]. To reach such colonization success, *B. schlosseri* evolved high resilience to changing physico-chemical conditions and resistance to polluted environments [25,26]. In fact, *B. schlosseri* is highly resilient to environmental changes [26] thanks to its ability to grow fast and to adjust the reproductive strategy within changes of environmental conditions. Consequently, it is considered one of the few species composing the benthic sessile community that is tolerant to highly polluted environments [25,27]. Finally, laboratory cultures of *B. schlosseri* [14] indicated this species as eurythermal and euryhaline, perfectly adapted to coastal environments [28].

Keeping in mind the peculiarities of the two models, we compared the effectiveness of sea urchins and golden star tunicates to reveal critical toxicity events due to heavy metals and organic compounds. Here, we propose formal methods for the correct assessment of environmental pollution in coastal seawater. We tested the effects of copper ions and polycyclic aromatic hydrocarbons (PAHs) on both organisms, and recorded their physiological responses at various doses. PAHs are a large class of compounds deriving from natural events (e.g., volcanic activity) and anthropogenic activities (e.g., industrial wastes, oil spills), affecting natural environments and associated biota [29,30]. In particular, 16 PAHs were defined as "highly toxic pollutants" by the U.S. EPA. Seven of them are considered highly carcinogenic by the International Agency for Research on Cancer [31–33]. Otherwise, copper is a typical transition metal, often found in coastal waters polluted by urban wastes. It is present in antifouling paints [34], having biocidal activity, but traces of Cu may derive from the steel industry [11] or from storm water runoffs [35] and, locally, from mining activities [36]. It is an essential element for marine organisms, in traces, but it is toxic at higher concentrations, prompting growth reduction [37], oxidative stress, and even sub-lethal and lethal effects [35,38] according to the doses and at concentrations lower than 0.25 mg L$^{-1}$. Acute toxicity effects on invertebrates are prompted at lower concentrations than the effects on fish and other vertebrates. Hence, Cu ions are often used in aquaculture to control fish pathologies provoked by invertebrate or protozoan parasites. Here, we aimed to mainly monitor acute effects prompted by Cu ions and sub-chronic effects prompted by PAHs, and evaluate the responses of both invertebrate models, in order to develop and optimize efficient ecotoxicity tests.

## 2. Materials and Methods

### 2.1. Ethics Statement for Paracentrotus lividus and Botryllus schlosseri

*Paracentrotus lividus* (Lamarck) and *Botryllus shlosseri* were collected from a site in the Bay of Naples that is not privately owned or protected in any way, according to Italian legislation (DPR 1639/68, 09/19/1980 confirmed on 01/10/2000). Field studies did not include endangered or protected species. All experimental procedures on animals were in compliance with the guidelines of the European Union (Directive 2010/63/EU).

### 2.2. Experimental Procedures for P. lividus

Adult sea urchins, *Paracentrotus lividus*, were collected by scuba divers in the Gulf of Naples (Figure 1) during the reproductive period and stored in tanks with circulating filtered seawater (FSW) until testing. Sea urchins were shaken to induce gamete emission or injected with 2M KCl through the peri-buccal membrane. Eggs were washed with FSW and kept refrigerated (4 °C) until experimental use. Dry sperm was collected and kept undiluted at 4 °C until use. Further, fertilized eggs and embryos were incubated in FSW added with PAHs and Cu at various concentrations (Table 1).

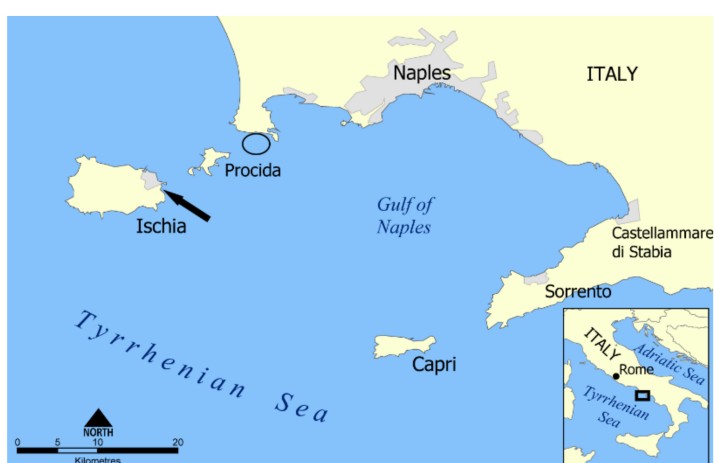

**Figure 1.** Sampling area in the gulf of Naples, Italy. The small circle indicates the sampling area for *P. lividus* off the mainland. The black arrow indicates the sampling station in the harbor of Ischia (40.745319N, 13.942339E) for *Botryllus schlosseri*.

**Table 1.** Concentrations of PAHs (µg/L) and Cu ions (mg/L) to which sea urchin embryos and *Botryllus schlosseri* colonies were exposed. The experiments started 10 min after the fertilization of sea urchin embryos or at the introduction of *B. shlosseri* colonies in multi-well plates.

| Concentrations | PAHs (µg/L) | Cu (mg/L) |
|---|---|---|
| A | 192 | 0.01 |
| B | $19.2 \times 10^2$ | 0.05 |
| C | $19.2 \times 10^3$ | 0.1 |
| D | $19.2 \times 10^4$ | 0.3 |
| E | $19.2 \times 10^5$ | 1 |
| F | - | 5 |

Experiments were performed in triplicate at each concentration of PAHs and Cu, using eggs from three different females. Control experiments tested fertilized embryos in FSW without contaminants. The embryonic development of sea urchins was followed over 48 h post-fertilization (hpf), up to the pluteus larval stage. Larvae were then fixed in glutaraldehyde (0.5% in FSW) and observed under an inverted microscope (Leica DMi6000 B) to detect the percentage of malformed plutei as compared to controls.

### 2.3. Experimental procedures for Botryllus schlosseri

*Botryllus schlosseri* were collected in the bay of Naples, harbor of Ischia (40°44′43″ N, 13°56′31″ E. Figure 1) in November and December 2019, when their natural populations were relatively abundant. Colonies were gently detached from submerged ropes and buoys using a razor blade and a paddle, and then carried to the laboratory. They were scrubbed, to detach encrusting algae and sediments, and individually positioned on glass slides prior to being moved to a 60 L tank containing 0.22 μm FSW. The tank, equipped with an air-stone, was kept in a thermostatic chamber at 17–18 °C, with a 12:12 h photoperiod, covered with a transparent lid to reduce evaporation, and water was replaced every two days. Colonies were fed once a day on axenic cultures of *Isochrysis galbana*. Glass slides and colonies were cleaned twice a week using a soft brush and a razor blade, in order to remove fecal pellets and encrusting algae or biofilms possibly influencing the physiology of filter-feeders [14].

Given the well-known physiology of adult blastozooids [39], toxic compounds such as Cu and PAHs were expected to trigger an anticipated apoptosis, as well as concentration-dependent death rates of colonies. To this end, young colonies (containing 5–10 adult zooids) were obtained from larvae produced by mother-colonies collected in the field and reared in the laboratory. In vitro tests lasted 48 h.

### 2.4. Collection Procedures

To obtain young colonies of *B. schlosseri* for toxicological tests, sexually mature colonies were reared in a flat tank, whose walls were covered with acetate transparent sheets, above which the newly hatched larvae settled and metamorphosed into the first oozoids. The transparent acetate sheets carrying the newly metamorphosed oozoids were transferred to another tank filled with 0.22 μm filtered seawater and kept in a vertical position using plastic pins [14]. The protocol described above for feeding, cleaning, and managing the culture tank was applied to the culture of oozoids and young colonies.

### 2.5. Toxicity Tests on Tunicates

Rapid screening tests were devised to evaluate the effects of two different allochemicals on the survival and recovery of colonial tunicates. To this end, colonies composed of 5–10 blastozooids were transferred from the tank into 6-well plates to obtain bioassays. Small fragments of transparent acetate sheets carrying experimental colonies were cut out and gently fixed to the bottom of the wells (one colony in each well) after polishing the colonies with a soft brush. Colonies were exposed to two treatments. Solutions were prepared, for each tested compound, at different doses (Table 1) and 10 mL of each treatment solution was poured in each well. Five solutions of PAHs were prepared by withdrawing aliquots of pure compound, to obtain five final concentrations of: 192, $19.2 \times 10^2$, $19.2 \times 10^3$, $19.2 \times 10^4$, and $19.2 \times 10^5$ μg/L (Table 1). For each treatment (including filtered seawater controls), 6 replicates were tested (36 colonies in total). In addition, six solutions of copper were prepared by diluting aliquots of pure compound ($CuCl_2$), to obtain final concentrations of: 0.01, 0.05, 0.1, 0.3, 1.0, and 5.0 mg/L. For each treatment (plus FSW control), 6 replicates were assayed, testing a total of 42 colonies.

Colonies were reared for 48 h in the wells. Mortality rates and health conditions of colonies were recorded by means of periodical observations under a Leica MZ6 optical microscope, at four times: Immediately at the start of the tests ($t_0$), and after 6 ($t_6$), 24 ($t_{24}$), and 48 h ($t_{48}$). A score code ranging from 1 to 5 was adopted to classify the health status of colonies (Table 2), obtaining an average value for each concentration under each treatment [28]. Records of pH, salinity, ammonium ($NH_3$ -N) concentration, and nitrites ($NO_2^-$ -N) were measured at the start ($t_0$) and at the end ($t_{48}$) of each test in the control solution using a pH-meter, a digital refractometer, and a Hach DR3900 spectrophotometer. Colonies were photographed under a Leica Z16 APO macroscope equipped with a Leica digital camera.

**Table 2.** Scores from 1 to 5 indicate the health state of colonies. The aspect and morphology of zooids and ampullae, pigmentation degree, appearance of circulatory systems, and transparency of the tunic were classified to record the health conditions of colonies [14].

| Scores | Aspect and Morphology |
|---|---|
| 1 | Zooids in healthy state, transparent zooids and ampullae, vessels in normal shape, open siphons, good hemolymph circulation |
| 2 | Zooids in healthy state but slightly pigmented and covered by microalgae, some closed oral siphons, not active filtering |
| 3 | Zooids and ampullae pigmented, some suffering zooids partially detached from the center, thick tunic, but normal vessels and ampullae normally distant from the colony |
| 4 | Zooids, ampullae, and vessels very dark and strict, zooids detached from the center and suffering, small and black, ruined shape, closed oral siphons. Peripheral circulation is limited, heartbeat (bpm) slows, and central circulation keeps the colony alive. |
| 5 | Zooids are dead, no heartbeat, and no circulation. Colony is darkish. |

*2.6. Statistical Analyses*

Health scores, morphological measures, and survival rates were reported as means $\pm$ standard deviation (SD). Data were analyzed by Shapiro–Wilk's test for normality ($p < 0.05$). The statistical significance of differences among treatments was checked by one-way ANOVA followed by Dunnett's test (GraphPad Prism Software version 8.02 for Windows, GraphPad Software, La Jolla, CA, USA, www.graphpad.com, accessed on 1 Februay 2021) for multiple comparisons. Differences were considered significant at $p < 0.05$.

**3. Results**

*3.1. Morphological Effects of PAHs and Cu on Sea Urchin Embryos*

Morphological observations at 48 hpf showed an increase in malformed and/or delayed plutei exposed to PAHs with respect to the control (Figure 2). In particular, at 192 and $19.2 \times 10^2$ μg/L, PAHs produced a significant increase in malformed plutei (16.3% at $p < 0.01$ and 16.4% at $p < 0.05$, respectively) with respect to controls (7.5%).

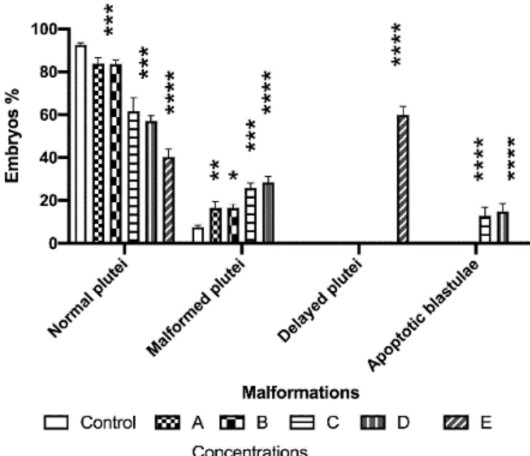

**Figure 2.** Percentage of normal plutei, malformed plutei, delayed plutei, and apoptotic blastulae produced by exposing newly fertilized eggs to five PAHs concentrations (A, 192, B, $19.2 \times 10^2$, C, $19.2 \times 10^3$, D, $19.2 \times 10^4$, E, $19.2 \times 10^5$ μg/L) in comparison with the control, represented by newly fertilized eggs in filtered sea water. Data are reported as mean $\pm$ standard deviation. The statistical significance of differences between test groups and controls is reported as follows: * $p < 0.05$, ** $p < 0.01$, *** $p < 0.001$, **** $p < 0.0001$.

Malformations of plutei mainly affected the apex, which appeared crossed, with arms that were broader (Figure 3E) or completely degenerated, and the entire body plan of the plutei strongly compromised and malformed (Figure 3F,G), compared to controls (Figure 3A). Increasing the concentrations of $19.2 \times 10^3$ and $19.2 \times 10^4$ µg/L, PAHs also induced an increase in malformed embryos (about 25.6%, $p < 0.0001$; and 28.3%, $p < 0.0001$, respectively); in these cases, about 12.8% and 14.7% (respectively) of embryos were still in the blastula stage ($p < 0.0001$). They showed evident apoptotic processes, while mitotic divisions occurred in the blastula, but their phenotypes were irregular (Figure 3C) and darker (Figure 3D).

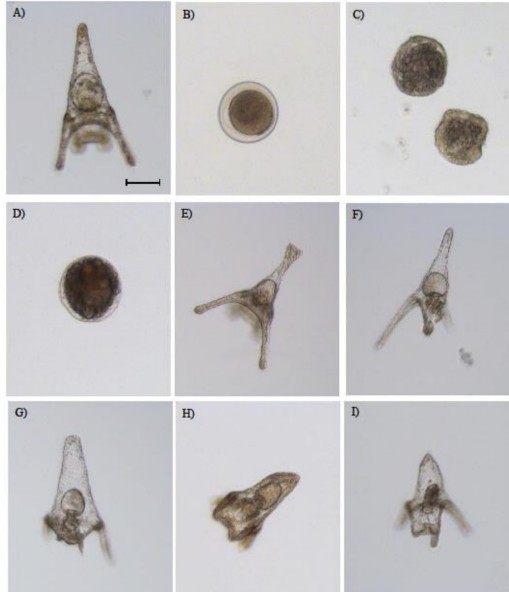

**Figure 3.** Examples of (**B**) fertilized eggs, (**C**,**D**) malformed blastulae, and (**E–I**) malformed plutei after exposure of fertilized eggs of *P. lividus* to PAHs and Cu, with respect to (**A**) the control, embryos deriving from fertilized eggs grown without PAHs and Cu. Scale bar = 50 µm; Leica DMi6000 B inverted microscope.

At the highest concentration ($19.2 \times 10^5$ µg/L), about 60% of embryos were delayed plutei ($p < 0.0001$). In fact, these embryos showed a delay in their development, revealed by a shortening of apex and arms (Figure 3H,I), with their morphology closely resembling that of controls, with a slight reduction in body length.

The exposure to 0.01 mg/L of Cu induced a significant increase in malformed plutei, (about 28%, $p < 0.0001$), with respect to the control (Figure 4). Furthermore, at increasing concentrations of 0.05 and 0.1 mg/L, all embryos appeared delayed still at the blastula stage ($p < 0.0001$), also being malformed (as reported in Figure 3C). At a concentration of 0.3 mg/L, Cu also induced a delay in embryonic development with about 52% of embryos still at the blastula stage (malformed). This also influenced the first mitotic division because 48% of fertilized eggs were prevented from overcoming the first mitotic division ($p < 0.0001$; see Figure 3B). At 1.0 mg/L, the Cu treatment showed a small percentage of malformed blastulae (about 6%, $p < 0.05$) and about 96% ($p < 0.0001$) of undivided eggs (Figure 4). At the highest concentration tested (5.0 mg/L), Cu blocked the first mitotic division of all the fertilized eggs ($p < 0.0001$).

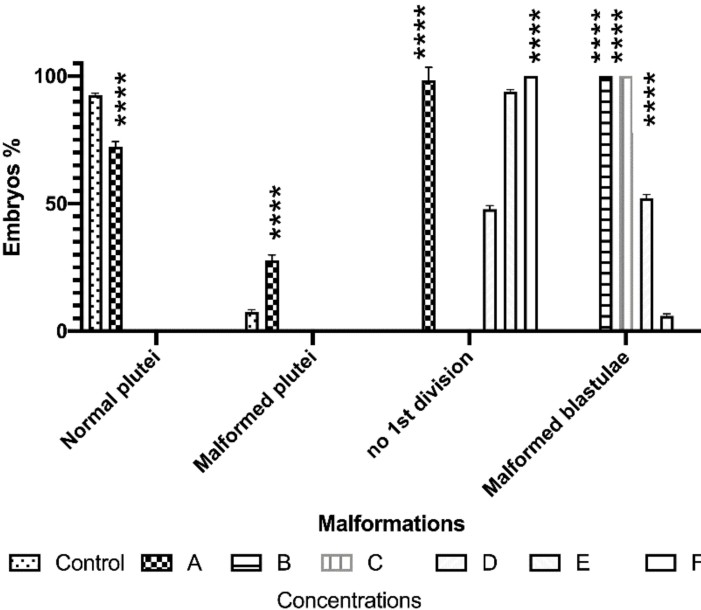

**Figure 4.** *P. lividus* exposed to Cu. Percentage of normal plutei, malformed, plutei, and eggs with arrest of the first mitotic division. Malformed blastulae were produced when newly fertilized eggs were exposed to Cu at the six concentrations tested (A, 0.01; B, 0.05; C, 0.1; D, 0.3; E, 1.0; and F, 5.0 mg/L) in comparison to controls, represented by newly fertilized eggs tested in FSW. The statistical significance of differences between test groups and controls is reported as follows: **** $p < 0.0001$).

### 3.2. Morphological Effects of PAHs and Cu on Botryllus shlosseri

#### 3.2.1. Chemical Descriptors of Media

Measures of physical and chemical features of the medium (Table 3) indicated an increase in salinity from 38 at $t_0$ to 40 PSU at $t_{48}$ and a decrease in pH from $8.14 \pm 0.01$ to $8.02 \pm 0.06$ in the same time range. Nitrites decreased by half, from $0.008 \pm 0.002$ ($t_0$) to $0.004 \pm 0.001$ ($t_{48}$), while ammonium increased significantly, from $0.06 \pm 0.012$ ($t_0$) to $0.76 \pm 0.84$ ($t_{48}$).

**Table 3.** Average ($\pm$ s.d.) of salinity, pH, $NO_2^-$ –N (mg/L), and $NH_3$ –N (mg/L) measured at the start of tests ($t_0$) and after 48 h ($t_{48}$).

| Time | Salinity (PSU) | pH | $NO_2^-$ -N (mg/L) | $NH_3$ -N (mg/L) |
|---|---|---|---|---|
| $t_0$ | $38.0 \pm 0.0$ | $8.14 \pm 0.01$ | $0.008 \pm 0.002$ | $0.06 \pm 0.012$ |
| $t_{48}$ | $40.0 \pm 0.0$ | $8.02 \pm 0.06$ | $0.004 \pm 0.001$ | $0.76 \pm 0.84$ |

#### 3.2.2. Development Times of Clutches

A range of development times was recorded at 18 °C, from the metamorphosis of the tadpole-like larva (first oozoid) up to the production of a colony made of several blastozooids (Figure 5). In four weeks, 53.1% of total metamorphosed oozoids formed colonies containing 8–10 blastozooids; 27.1% formed colonies containing 4–6 blastozooids; 9.4% formed colonies containing 2–3 blastozooids. However, the remaining 3.1% oozoids developed into a single bud, neither doubling nor producing a growing colony. In addition, a small percentage of oozoids (1.04%) produced a colony containing more than 12 blasto-zooids (and up to 16) in 4 weeks. The total percentage of dead oozoids (either immediately after metamorphosis or after developing in the first blastozooid) not producing colonies was 6.3%. Accordingly, the largest frequency classes, i.e., those containing colonies of 4–6 and 8–10 zooids, were used for in vitro tests of pollutants.

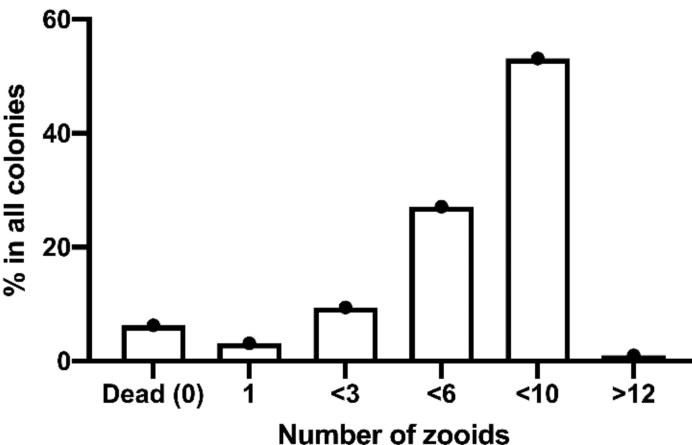

**Figure 5.** Percentage of colonies of *Botryllus shlosseri* containing different numbers of blastozooids at the fourth week after settlement.

### 3.2.3. Effects of PAHs

The administration of PAHs did not trigger stress signals or lethal effects on *B. schlosseri* colonies. Colonies treated with 192, $19.2 \times 10^2$, $19.2 \times 10^3$, and $19.2 \times 10^4$ μg/L remained in their best state (score 1) throughout the experiment, up to 48 h of exposure. During this exposure time, they were visually comparable to the controls according to pigmentation, heartbeat rhythms (bpm), hemolymph circulation, distance between the zooids and ampullae, transparency of the matrix, and feeding activities (Figure 6a,b). Only at the maximum dose ($19.2 \times 10^5$ μg/L) were 60% of treated colonies shifted to a score 2, at $t_{24}$. In parallel, their heartbeat decreased from $95.60 \pm 21.65$ ($t_0$) to $58.80 \pm 16.28$ bpm ($t_{24}$). Similarly, at $t_{48}$ (Figure 6c), they exhibited a few closed (not-filtering) oral siphons and a darker pigmentation of ampullae, shifting to a health score 2. However, all colonies survived in the range of concentrations of PAHs tested, over 48 h of exposure, and slight signals of stress were exhibited at $t_{24}$ and $t_{48}$ only at the maximum concentration.

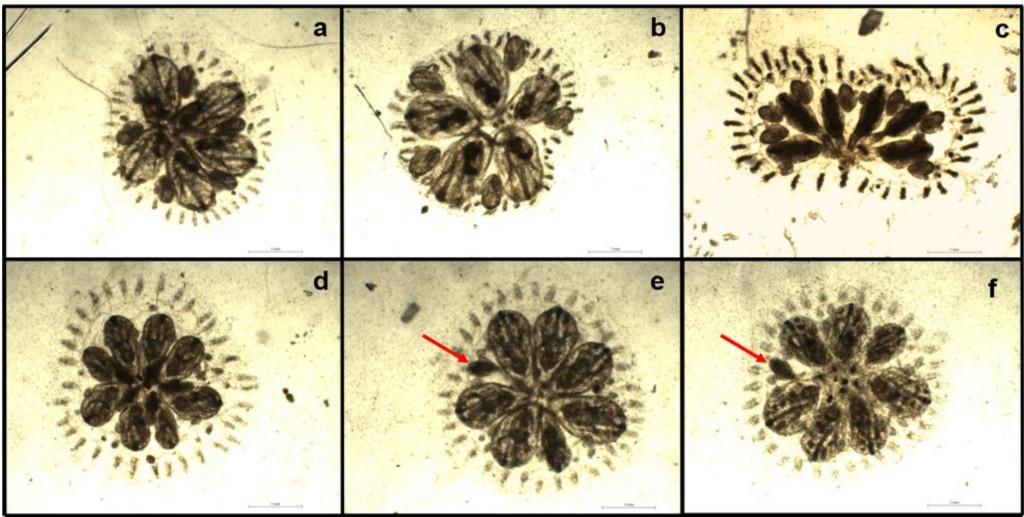

**Figure 6.** Colonies of *B. schlosseri* treated with PAHs to show a range of possible morphological reactions and their classification according to health scores. (**a**) Control at $t_{24}$; (**b**) concentration $19.2 \times 10^5$ μg/L at $t_{24}$, showing a score 1 colony; (**c**) concentration $19.2 \times 10^5$ μg/L at $t_{48}$, showing a score 3 colony; (**d**) concentration $19.2 \times 10^4$ μg/L at $t_{24}$; (**e**) concentration $19.2 \times 10^4$ μg/L at $t_{48}$ showing most zooids in the good state (score 1) and one malformed zooid (red arrow); (**f**) same colony as in (**e**) observed by the opposite site of the acetate. Both (**e**) and (**f**) show a zooid in a phase of re-absorption (red arrow).

### 3.2.4. Effect of Copper

The administration of copper to *B. schlosseri* colonies induced dose-dependent effects. Health conditions of animals (as referred to scores indicated in Table 2) were stable within the duration of the experiment at three lower doses (0.01, 0.05, and 0.1 mg/L). The lower doses did not trigger negative effects within 48 h (Figure 7) and the state of health of colonies was comparable to that of control colonies (score 2). However, a progressive deterioration was observed at higher concentrations. In fact (Figure 8b), all specimens treated at the highest concentration (5.0 mg/L) died. In particular, the administration of 0.3 mg/L of Cu induced stress (score 3) at $t_{24}$ and $t_{48}$ (Figure 8a–d).

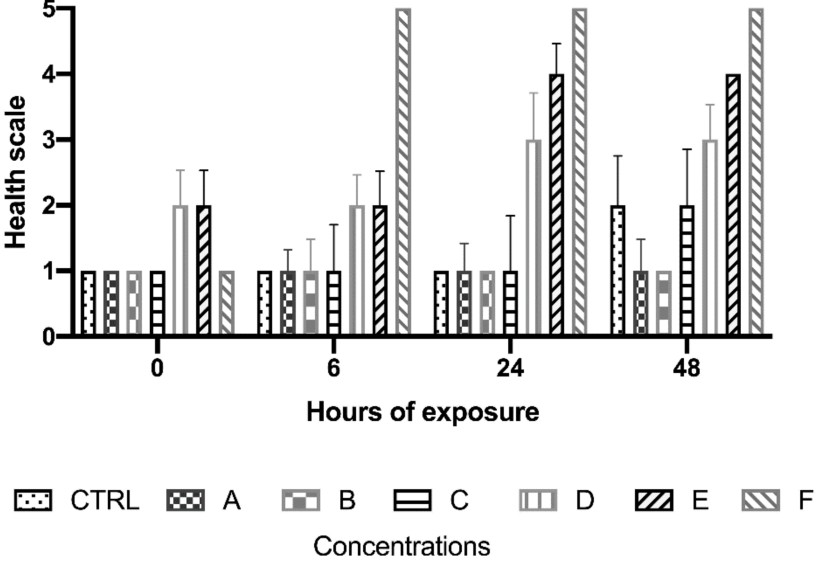

**Figure 7.** Effect of the exposure of *Botryllus shlosseri* to copper at 6 concentrations, as compared to controls, checked at four time-intervals (from $t_0$ to $t_{48}$). The patterns of bars indicate concentrations. Scores from 1 to 5 on the abscissa are referred to health classes indicated in Table 2.

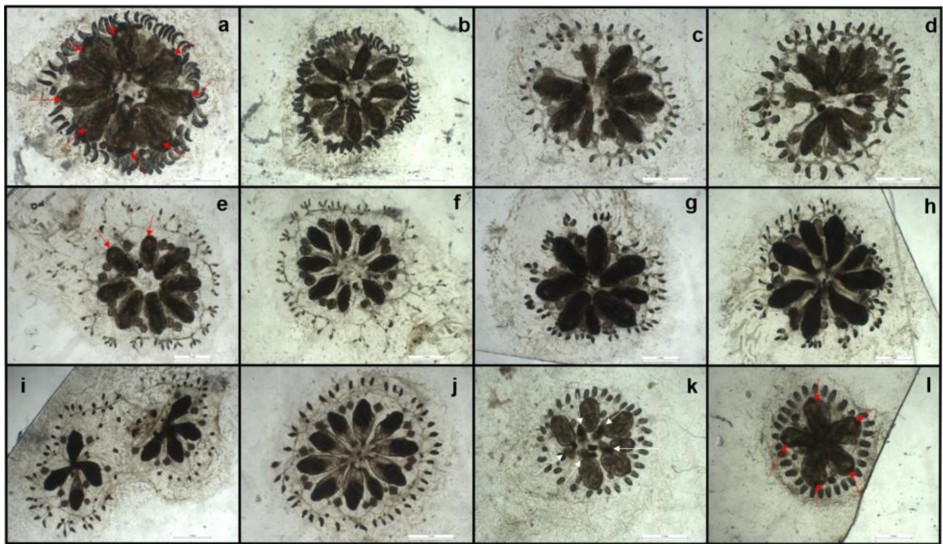

**Figure 8.** Colonies of *B. schlosseri* exposed to Cu ions. Colonies exposed to 0.3 mg/L at $t_{24}$ (**a**) and at $t_{48}$ (**b**); colonies exposed to 0.3 mg/L at $t_{24}$ (**c**) and at $t_{48}$ (**d**); colonies exposed to 1.0 mg/L at $t_{24}$ (**e**) and at $t_{48}$ (**f**); colonies exposed to 1.0 mg/L at $t_{24}$ (**g**) and at $t_{48}$ (**h**); colonies exposed to 5.0 mg/L at $t_{24}$ (**i**); colonies exposed to 5.0 mg/L at $t_{24}$ (**j**); control colonies (pure medium, without Cu) under take-over at $t_{48}$ (**k,l**). White arrows in panel (**k**) indicate reabsorbed adults. Red arrows in (**a,l**) indicate open and actively filtering inhalant siphons. White scale bars: 1 mm.

Zooids exhibited a partial detachment of the atrial siphons, starting from the common atrial channel, and a darker pigmentation also involving their ampullae. The zooid matrix, which was almost transparent in normal individuals, became thicker and darker, but colonies still exhibited the typical shape of vessels and elongated ampullae. Colonies treated with 1.0 mg/L of Cu (Figure 8e–h) were classified with score 3 at $t_{24}$, and score 4 at $t_{48}$. Zooids, vessels, and ampullae were dark and thin, zooids were disconnected from the center of the colony, and the common atrial aperture was lost. The circulation of hemolymph in the periphery of the colony was reduced. Ampullae contained several pigmented cells conferring them a dark aspect. The inhaling oral siphons of all zooids were sealed and not visible. These features dramatically increased between $t_{24}$ and $t_{48}$. Colonies under the treatment at the maximum concentration (5 mg/L) died after 6 h (Figure 8i,j). No heartbeat nor circulation were recorded; zooids, ampullae, and vessels appeared black. However, during the experiment, a few colonies underwent a natural take-over process and a darker color was consequently exhibited (Figure 8k,l) but normal shapes and positions of ampullae permitted the distinguishing of this process. In fact, after the take-over, buds were replaced and new budlets were further produced. By contrast, the deterioration of colonies was completed in 48 h at lower concentrations (0.3 and 1.0 mg/L), while the highest concentration (5.0 mg/L) led colonies to death in 6 h.

## 4. Discussion

The results obtained indicate that the two models answer differently to possible environmental stressors. Sea urchin embryos exhibit high sensitivity and responses dependent on the concentration of pollutants, as already indicated by previous research [40,41]. This marine invertebrate has a limiting constraint, represented by the seasonal availability of embryos, generally limited to colder months (although other sea urchins may also be available in warmer seasons). In parallel, this study confirms that botryllids may represent excellent models to investigate the presence and the effects of allochemicals in marine environments, supporting previous studies, which indicate the power of various tunicates as good indicators of environmental health [16]. Botryllid may be easily cultured in the laboratory in semi-automatic conditions [42] and remain available in all seasons, if correctly managed. In addition, their young colonies may easily be transferred into multi-wells using the methods described herein, and immediately exposed to the contaminants. We demonstrated that obtaining colonies with an optimal numeric abundance of zooids (between 4 and 10) is easy, permitting immediate testing in the laboratory conditions. In fact, as demonstrated by our laboratory tests, *B. schlosseri* was not affected by variations in physical and chemical features of water, and all growing colonies exhibited a natural blastogenetic cycle. Although data on the tolerance of *B. schlosseri* to nitrites and nitrates are still missing, the diffusion of this species in coastal zones (where river outflows discharge large amounts of ammonium forms in the seawater) indicates its resistance to nitrite and nitrate pollution.

Our tests on the effect of such different xenobiotics as PAHs and heavy metals showed clear differences between the two models. Sea urchin embryos indicated a clear dose-dependence sensitivity to PAHs. The percentage of normal plutei, in particular, linearly decreased according to the concentrations of PAHs, and in parallel, the amount of malformed plutei increased. These results confirmed the perfect fitting of this model to identify the presence of compounds possibly producing chronic effects in the marine biota. In the case of sea urchins, both PAHs and Cu demonstrated toxigenic effects at the experimental concentrations tested. PAHs exhibited lower toxicity with respect to Cu. In fact, PAHs revealed a dose-dependent effect in producing malformed embryos, up to the highest concentrations at which they induced a delay in embryonic development. In the case of Cu, a very strong delay of *P. lividus* embryonic development was observed with a total block of the first mitotic division.

Several studies were performed to detect the possible impacts of single compounds or solutions of PAHs on sea urchin development. Suzuki et al. [43] showed that PAHs were

able to influence spicule formation of *Hemicentrotus pulcherrimus* during embryogenesis, inhibiting the expression of vascular endothelial growth factor (VEGF) [44]. Pillai et al. [45] demonstrated that five PAHs (phenanthrene, fluoranthene, fluorene, pyrene, and quinoline) affected the axial development and patterning in embryos of the sea urchin *Lytechinus anemesis*, by disrupting the regulation of β-catenin. The dangerous impact of Cu was thus predictable.

The effects of heavy metals on sea urchins were also extensively studied. Kobayashi and Okamura [46] evaluated the effect of eight metals (manganese, lead, cadmium, nickel, zinc, chromium, iron, and copper) on the embryo development of the sea urchin *Anthocidaris crassispina*, demonstrating that Cu was the most toxic element, with a "no observed effect concentration" (NOEC) of 3.8 μg/L. Bielmyer et al. [47] exposed embryos of *Diadema antillarum* to dissolved copper, detecting an LC50 of 25 μg/L. His et al. [48] tested the sensitivity of the oyster *Crassostrea gigas* and the sea urchin *P. lividus* embryos and larvae to various contaminants (including Cu), showing that *Crassostrea gigas* was more sensitive to Cu than *P. lividus*.

By contrast, *B. shlosseri* did not react significantly to our range of concentrations of PAHs, with the only exception of the highest concentrations, at which it demonstrated an effect. These results may be explained taking into account its well-known tolerance to PAHs. After their release in the marine environment, PAHs are retained in marine sediments, altering the benthic biocenosis [4]. For example, PAHs trigger a decrement in the growth rates of the amphipod *Leptocheirus plumulosus* [49], and the behavior of young colonies of botryllids is similar to that of adult ones, as regards to survival and quality of life, with the exception that the optimal area of adult colonies is wider than that of young colonies [28]. However, despite the strong tolerance to PAHs and changes in environmental parameters, *B. schlosseri* has been proven to be sensitive to biocides, and in particular, to toxic compounds contained in antifouling paints, widely used on the submerged part of boats, ships, and artificial structures [13,17,50–52]. In fact, its responses to the presence of heavy metals were clearer and proportional to the concentration, especially after 24 and 48 h of exposure. Consequently, although various authors indicated *B. schlosseri* as a good bioindicator for ecotoxicological studies on organic xenobiotics [53,54], this study demonstrates the limits for such applications. Many studies were conducted on the toxicity of PAHs on marine organisms of commercial and ecological importance [5,55,56]. Bellas et al. [57] performed toxicity bioassays on embryos and larval stages of the sea urchin *Paracentrotus lividus*, the ascidian *Ciona robusta*, and the mussel *Mytilus galloprovincialis*, by exposing them to five potentially toxic PAHs. They demonstrated that all five PAHs inhibited and reduced the larval development and the growth of both *M. galloprovincialis* and *P. lividus*. The embryos and larval stages of *C. intestinalis* showed to be more tolerant and suffered toxic effects only by one PAH (naphthalene). Bellas and Thor [58] demonstrated the high sensitivity of the copepod *Acartia tonsa*, particularly regarding reproduction and recruitment. The two models proposed summarize the responses provided by such different organisms.

In conclusion, the two invertebrate models may provide complementary information on dissolved allochemicals, and their reactions are in line with their specific sensitivity. PAHs must be considered insidious compounds, because their chronic effects are not easily detected in marine invertebrates, especially in early phases, although they are clearly demonstrated in polluted communities. Sea urchin embryos may detect even low concentrations of PAH due to their specific sensitivity, while the effects on botryllid colonies are probably to be expected in longer time periods, due to progressive bioaccumulation. By contrast, heavy metals produce acute toxigenic effects on both models and, in this view, botryllid may offer even clearer responses, showing a dose-dependent range of reactions.

From an ecological point of view, various impacts, such as pollution and species invasions [59], may potentially interact and trigger [60] decreases in biodiversity [61,62]. The interaction between anthropogenic stressors and invaders themselves could significantly influence species colonization, distribution, and survival in various environments [63]. The Invader Tolerance Model (ITM) developed by Osborne et al. [64] provides a framework to

explain the reduction in biodiversity by invasive species and marine pollution, and it proposes that invaders are more tolerant to pollution relative to native species. Consequently, according to this model, it is forecasted that native species diversity will decrease following an exposure event. This decrease will further enable invaders to settle, thus producing a loop of declining diversity. However, within the ITM, there are unanswered questions regarding the relative importance of pollution and invasive species on native species, as well as their combined effects. The implementation of such different models as planktonic organisms (sea urchin larvae) and benthic filter feeders (Botryllid colonies) will help to investigate these hypotheses.

**Author Contributions:** Conceptualization, V.Z. and M.C.; methodology, V.Z. and M.C.; software, V.Z. and E.S.; formal analysis, V.Z. and M.C.; investigation, C.G., E.S. and L.A.; resources, V.Z. and M.C.; data curation, C.G., E.S. and L.A.; writing—original draft preparation, V.Z., C.G. and L.A.; writing—review and editing, V.Z. and M.C.; supervision, V.Z. and M.C.; project administration, V.Z.; funding acquisition, M.C. and V.Z. All authors have read and agreed to the published version of the manuscript.

**Funding:** This research was funded by ABBACO, grant number CIPE-GU n. 56 8.3.2017).

**Institutional Review Board Statement:** Not applicable

**Informed Consent Statement:** Not applicable

**Data Availability Statement:** The data presented in this study are available on request from the corresponding author.

**Acknowledgments:** Luisa Albarano was supported by a PhD (PhD in Biology, University of Naples Federico II) fellowship co-funded by the Stazione Zoologica Anton Dohrn and University of Naples Federico II. We acknowledge the Fishery Service for providing sea urchins and Davide Caramiello of the Marine Resources for Research Unit of the Stazione Zoologica for his technical support for maintenance and gamete collection. English text was kindly revised by R. Messina, of the Stazione Zoologica Anton Dohrn.

**Conflicts of Interest:** The authors declare no conflict of interest.

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
