# Peer review of "Assessing the Ecotoxicity of Copper and Polycyclic Aromatic Hydrocarbons: Comparison of Effects on Paracentrotus lividus and Botryllus schlosseri, as Alternative Bioassay Methods"

_water, doi:10.3390/w13050711_

Round 1
Reviewer 1 Report
The reviewed study entitled Assessing of the ecotoxicity of Copper and Polycyclic aromatic hydrocarbons: comparison of effects on Paracentrotus lividus and Botryllus schlosseri, as alternative bioassay methods concerns a very important problem of organic and inorganic pollutants of environment. Rapid assessment of a heavy metals and Polycyclic Aromatic Hydrocarbons contamination is important. The rapid assessment helps to reduce a range and effects of pollution of the water environment.
Author Response
Reviewer 1
Comments and Suggestions for Authors
The reviewed study entitled Assessing of the ecotoxicity of Copper and Polycyclic aromatic hydrocarbons: comparison of effects on Paracentrotus lividus and Botryllus schlosseri, as alternative bioassay methods concerns a very important problem of organic and inorganic pollutants of environment. Rapid assessment of a heavy metals and Polycyclic Aromatic Hydrocarbons contamination is important. The rapid assessment helps to reduce a range and effects of pollution of the water environment.
We thank the reviewer for the positive evaluation of our manuscript.
Reviewer 2 Report
The presented manuscript is interesting. The search for bioindicators is not an original issue, but I believe that the work provides new information on the subject and may be published in the journal Water. I would suggest Authors to emphasize the novelty of their work in relation to existing literature.
Please check references:
no 7 (no reference in the introduction – Line 43);
no 33 (Introduction - Line 89)
I have no more comments regarding the article.
Author Response
Reviewer 2
Comments and Suggestions for Authors
The presented manuscript is interesting. The search for bioindicators is not an original issue, but I believe that the work provides new information on the subject and may be published in the journal Water. I would suggest Authors to emphasize the novelty of their work in relation to existing literature.
We thank the reviewer for the accurate evaluation of our manuscript.
Please check references:
no 7 (no reference in the introduction – Line 43);
The error was corrected
no 33 (Introduction - Line 89)
The error was corrected
I have no more comments regarding the article.
Reviewer 3 Report
The authors evaluated the relative toxicity of two common marine pollutants, polycyclic aromatic hydrocarbons (PAHs) and copper, to two species of marine invertebrates Paracentrotus lividus and Botryllus schlosseri to aid in establishing standardized toxicity testing methods. Considering the ecological relevance of these two species and the relative differences in expected toxicity based on life history characteristics, better understanding the effects of PAHs and copper on these two species provides valuable insight into the effects of anthropogenic pollution on marine ecosystems. As expected, P. lividus showed sensitivity to both PAHs and copper in a dose dependent manner. Toxic effects resulting from exposure to these common environmental pollutants was evaluated through morphological observations made during embryonic development. B. schlosseri showed more resistance to PAHs than that of P. lividus but was extremely sensitive to copper. Considering the relative ease of maintaining cultures of these two species under laboratory conditions and the easily definable endpoints related to toxicity, the authors present a compelling case regarding the potential use of these two species as test species in environmental toxicology. Similarly, the authors provide a comprehensive set of pictures describing the morphological changes used to indicate toxic effects of exposure, which will allow other scientists to utilize these species in toxicity tests. Overall, the authors provide a comprehensive analysis of the methods required for utilizing P. lividus and B. schlosseri in toxicity testing and the relative sensitivity of these two species to common marine contaminants. Therefore, it is recommended that the manuscript be accepted for publication. There are some grammatical errors throughout the entire manuscript that will require revision. Some specific examples are included below.
Specific Comments
Line 17: “as organic and inorganic pollutants” should be “such as organic and inorganic pollutants”.
Line 22: “expositions” should be “exposures”.
Line 78: “costal” should be “coastal”.
Line 81: The comma between “compounds” and “to” should be removed.
Line 87: “PAHs were defined ‘highly toxic pollutants’” should be “PAHs were defined as ‘highly toxic pollutants’”.
Line 172: “essayed” should be “assayed”.
Line 233: “were prevented to overcome” should be “were prevented from overcoming”.
Table 2: The commas in the numbers should be replaced with periods.
Line 279: The units for heartbeat should be included.
Line 316: “dead yet” should be “died”
Author Response
Reviewer 3
Comments and Suggestions for Authors
The authors evaluated the relative toxicity of two common marine pollutants, polycyclic aromatic hydrocarbons (PAHs) and copper, to two species of marine invertebrates Paracentrotus lividusand Botryllus schlosseri to aid in establishing standardized toxicity testing methods. Considering the ecological relevance of these two species and the relative differences in expected toxicity based on life history characteristics, better understanding the effects of PAHs and copper on these two species provides valuable insight into the effects of anthropogenic pollution on marine ecosystems. As expected, P. lividus showed sensitivity to both PAHs and copper in a dose dependent manner. Toxic effects resulting from exposure to these common environmental pollutants was evaluated through morphological observations made during embryonic development. B. schlosseri showed more resistance to PAHs than that of P. lividus but was extremely sensitive to copper. Considering the relative ease of maintaining cultures of these two species under laboratory conditions and the easily definable endpoints related to toxicity, the authors present a compelling case regarding the potential use of these two species as test species in environmental toxicology. Similarly, the authors provide a comprehensive set of pictures describing the morphological changes used to indicate toxic effects of exposure, which will allow other scientists to utilize these species in toxicity tests. Overall, the authors provide a comprehensive analysis of the methods required for utilizing P. lividus and B. schlosseri in toxicity testing and the relative sensitivity of these two species to common marine contaminants. Therefore, it is recommended that the manuscript be accepted for publication. There are some grammatical errors throughout the entire manuscript that will require revision. Some specific examples are included below.
We thank the reviewer for the positive and accurate evaluation. We have followed all the useful suggestions contained in “specific comments” and, in addition, we have revised once again all the text, with the aid of a mother-language colleague, to improve the quality of the manuscript as indicated by the referee.
Specific Comments
Line 17: “as organic and inorganic pollutants” should be “such as organic and inorganic pollutants”.
Done as suggested
Line 22: “expositions” should be “exposures”.
Done
Line 78: “costal” should be “coastal”.
Done
Line 81: The comma between “compounds” and “to” should be removed.
Done
Line 87: “PAHs were defined ‘highly toxic pollutants’” should be “PAHs were defined as ‘highly toxic pollutants’”.
Done
Line 172: “essayed” should be “assayed”.
Done
Line 233: “were prevented to overcome” should be “were prevented from overcoming”.
Done
Table 2: The commas in the numbers should be replaced with periods.
Done
Line 279: The units for heartbeat should be included.
Done
Line 316: “dead yet” should be “died”
Done